# Differential Effects of Dietary versus Exercise Intervention on Intrahepatic MAIT Cells and Histological Features of NAFLD

**DOI:** 10.3390/nu14112198

**Published:** 2022-05-25

**Authors:** Sara Naimimohasses, Philip O’Gorman, Ciara Wright, Deirdre Ni Fhloinn, Dean Holden, Niall Conlon, Ann Monaghan, Megan Kennedy, John Gormley, Peter Beddy, Stephen Patrick Finn, Joanne Lysaght, Jacintha O’Sullivan, Margaret R. Dunne, Suzanne Norris, J. Bernadette Moore

**Affiliations:** 1Department of Clinical Medicine, Trinity College, 2 Dublin, Ireland; naimims@tcd.ie; 2Hepatology Department, St. James’s Hospital, 8 Dublin, Ireland; 3Discipline of Physiotherapy, Trinity College, 2 Dublin, Ireland; pogorma@tcd.ie (P.O.); anmonagh@tcd.ie (A.M.); kennedme@tcd.ie (M.K.); jgormley@tcd.ie (J.G.); 4Glenville Nutrition, 6 Dublin, Ireland; ciara.wright@glenvillenutrition.ie; 5Department of Nutritional Sciences, University of Surrey, Surrey GU2 7XH, UK; nifhlod@tcd.ie; 6Department of Immunology, St. James’s Hospital, 8 Dublin, Ireland; holdend@tcd.ie (D.H.); conlonn1@tcd.ie (N.C.); 7Department of Radiology, St. James’s Hospital, 8 Dublin, Ireland; pbeddy@stjames.ie; 8Department of Histopathology, St. James’s Hospital, 8 Dublin, Ireland; sfinn@stjames.ie; 9Department of Surgery, School of Medicine, Trinity College, 2 Dublin, Ireland; jlysaght@tcd.ie (J.L.); osullij4@tcd.ie (J.O.); 10Trinity Translational Medicine Institute, Trinity College, 2 Dublin, Ireland; margaret.a.dunne@gmail.com; 11Department of Applied Science, Technological University Dublin, 24 Dublin, Ireland; 12School of Food Sciences and Nutrition, University of Leeds, Leeds LS2 9JT, UK

**Keywords:** NAFLD, MAFLD, MAIT cells, diet, exercise, weight loss intervention

## Abstract

Background: Mucosal-associated invariant T (MAIT) cells promote inflammation in obesity and are implicated in the progression of non-alcoholic fatty liver disease (NAFLD). However, as the intrahepatic MAIT cell response to lifestyle intervention in NAFLD has not been investigated, this work aimed to examine circulating and intrahepatic MAIT cell populations in patients with NAFLD, after either 12 weeks of dietary intervention (DI) or aerobic exercise intervention (EI). Methods: Multicolour flow cytometry was used to immunophenotype circulating and intrahepatic MAIT cells and measure MAIT cell expression (median fluorescence intensity, MFI) of the activation marker CD69 and apoptotic marker CD95. Liver histology, clinical parameters, and MAIT cell populations were assessed at baseline (T0) and following completion (T1) of DI or EI. Results: Forty-five patients completed the study. DI participants showed decreased median (interquartile range) expression of the activation marker CD69 on circulating MAIT cells (T0: 104 (134) versus T1 27 (114) MFI; *p* = 0.0353) and improvements in histological steatosis grade post-intervention. EI participants showed increased expression of the apoptotic marker CD95, both in circulating (T0: 1549 (888) versus T1: 2563 (1371) MFI; *p* = 0.0043) and intrahepatic MAIT cells (T0: 2724 (862) versus T1: 3117 (1622) MFI; *p* = 0.0269). Moreover, the percentage of intrahepatic MAIT cells significantly decreased after EI (T0: 11.1 (14.4) versus T1: 5.3 (9.3)%; *p* = 0.0029), in conjunction with significant improvements in fibrosis stage and hepatocyte ballooning. Conclusions: These data demonstrate independent benefits from dietary and exercise intervention and suggest a role for intrahepatic MAIT cells in the observed histological improvements in NAFLD.

## 1. Introduction

Non-alcoholic fatty liver disease (NAFLD) is now the leading cause of chronic liver disease worldwide, with an estimated global prevalence of 25% [1]. The term NAFLD covers a broad spectrum of disease severity, that can progress from simple steatosis to non-alcoholic steatohepatitis (NASH) and hepatic fibrosis [2]. Progression to fibrosis is driven by lipotoxicity, oxidative stress, and mitochondrial dysfunction that leads to hepatocellular injury [3]. These cellular changes, in addition to the translocation of gut-derived pathogens, promote chemotaxis and the stimulation of innate and adaptive immune cells that drive fibrosis [4]. Hepatic fibrosis is the strongest prognostic factor for morbidity and mortality, and conveys high risks of cardiovascular disease as well as cirrhosis and hepatocellular carcinoma (HCC) [5]. More recently, NAFLD has emerged as a key risk factor for severe COVID-19. The chronic inflammation driven by oxidative stress associated with NAFLD attenuates the immune response to viral infections, resulting in higher viral loads and more rapid disease progression [6,7].

The integral relationship between NAFLD, obesity, and metabolic dysfunction has prompted a consensus-driven proposal both for a name change to ‘metabolic-associated fatty liver disease’ (MAFLD), and a shift from exclusionary diagnostic criteria [8,9]. If fully adopted, the proposed positive diagnostic criteria will be hepatic steatosis in the presence of either overweight/obesity or type 2 diabetes, or the presence of two metabolic risk factors in normal weight individuals [9]. Currently, there are no approved pharmacotherapies for NAFLD, and given the close association between NAFLD and obesity, weight loss through dietary and lifestyle intervention is central to current clinical management guidelines [10,11]. Lifestyle intervention trials focusing on weight loss have shown histological improvements in NAFLD [12], with similar results observed in patients with weight loss resulting from pharmacotherapy or bariatric surgery [13]. Resolution of NASH has been observed in individuals achieving 5% or more weight loss, and fibrosis regression noted in those who lost more than 10% of their initial bodyweight [14]. Interestingly, the combination of diet and exercise appears to result in greater improvements in markers of NAFLD activity than either intervention alone [12].

Mucosal-associated invariant T (MAIT) cells are “innate-like” T lymphocytes that possess features of both innate and adaptive immunity, and which are particularly enriched in the liver where they can comprise up to 45% of human intrahepatic lymphocytes [15]. MAIT cells are defined by the expression of an invariant Vα7.2 T cell receptor chain and high expression of the CD161 C-type lectin receptor, which is also expressed by natural killer cells [16]. Upon stimulation, subpopulations of MAIT cells rapidly produce different inflammatory cytokines such as Interferon γ (IFNγ), Tumor Necrosis Factor α (TNF α), and Interleukin-17 (IL-17). In particular, IL-17-producing MAIT cells have been implicated in multiple chronic inflammatory diseases, including obesity and NAFLD [17]. Circulating MAIT cells are markedly decreased in adults with obesity and/or type 2 diabetes, as well as children with obesity [18,19]. The remaining circulating MAIT cells, as well as MAIT cells in adipose tissue, were shown to have activated, inflammatory, and IL-17-producing phenotypes [18,19]. Moreover, following bariatric surgery with associated improvements in metabolic parameters [18], circulating MAIT cells increased in number and had decreased cytokine production.

The role of MAIT cells has only recently begun to be investigated in chronic liver disease, with few studies in humans reported to date. In patients with end-stage liver disease undergoing transplantation from a variety of causes including NASH, MAIT cells numbers were reduced in both blood and the explanted liver tissue [20]. Similarly, in patients with NAFLD, the levels of circulating MAIT cells are decreased in comparison to healthy controls [21,22,23]. Whether intrahepatic MAIT cells are protectors or protagonists in the context of NAFLD pathogenesis is not yet clear, and may depend on disease stage [16]. Recent work shows that liver resident MAIT cells have unique transcriptomic effector profiles and a more polyfunctional phenotype than their blood counterparts [24]. In animal studies, wild-type mice fed a methionine choline-deficient diet increased their intrahepatic MAIT cell number, and MAIT knock out mice had more severe hepatic steatosis and inflammation, implying a protective role in NAFLD-related inflammation [22]. However, in models of chronic liver injury, MAIT knock out mice exhibited less fibrosis indicating a profibrotic role for MAIT cells [21].

In situ staining of paraffin-embedded liver tissue from patients with NAFLD (*n* = 40) has suggested an increased number of MAIT cells in NAFLD in comparison to healthy control liver (*n* = 5) [22]. However, a study using flow cytometry and immunophenotyping demonstrated no difference in intrahepatic MAIT cell numbers in liver biopsies from patients with NAFLD (*n* = 15) versus healthy livers (*n* = 3) [23]. No studies to date have investigated changes in MAIT cells in the liver following lifestyle interventions in NAFLD. Our study aimed to examine circulating and intrahepatic MAIT cell populations and metabolic and histological changes in patients with NAFLD following either a 12-week dietary or aerobic exercise intervention.

## 2. Materials and Methods

### 2.1. Ethics

This feasibility study was approved by the St. James’s and the Tallaght University Hospitals, Dublin, Ireland, by the Research Ethics Committee; approval/registration: 2017-05 List 17(12). Written informed consent was obtained from all patients and the study was conducted in accordance with the guidelines outlined in the Declaration of Helsinki (2013). 

### 2.2. Participants

Fifty patients with biopsy confirmed NAFLD, attending the hepatology out-patient clinic at St James’s Hospital, Dublin, Ireland, were enrolled in the study. Inclusion criteria were as follows: age ≥18, biopsy-proven NAFLD, and the ability to attend either weekly nutrition or bi-weekly exercise classes in St James’s Hospital for 12 weeks. Exclusion criteria were as follows: contraindications to liver biopsy; exercise testing; significant orthopaedic or neuromuscular limitations; unwillingness to participate; alcohol consumption >30 g/day (males) or >20 g/day (females); co-existing liver disease; or treatment with medications associated with hepatic steatosis and/or chronic liver disease, e.g., Methotrexate, Amiodarone, Tamoxifen. 

### 2.3. Study Design

In this three-arm controlled pilot and feasibility study, following baseline assessments (T0), participants were allocated by convenience sampling to either a control group (CG, *n* = 16), a diet intervention (DI, *n* = 16), or an exercise intervention (EI, *n* = 18) for 12 weeks. Recruitment was limited by patient accessibility and availability to attend for intervention and follow up; therefore, participants were not randomized, nor was it possible to blind them to the interventions. While the CG received standard of care, including a recommendation of self-directed weight loss, the multi-component DI incorporated weekly group meetings focused on nutrition education and behavioural change, as well as weigh-ins and individualized support from trained nutritionists. Participants were asked to keep a weekly food diary that was advised on, on a one-to-one basis, and adherence to the recommendations were assessed at each weekly session. The diet promoted was moderately hypocaloric and included aspects of the Mediterranean diet. Consumption of foods with high fibre content, low glycaemic load carbohydrates, and replacement of saturated fat with mono- and polyunsaturated fats was emphasized. Recipes and food plans provided centred on increased whole foods, fish, nuts/seeds, legumes, vegetables, and complex carbohydrates, and aimed to reduce reliance on meat and processed foods. No supplements were recommended for the duration of the study. Overall dietary guidelines were provided to patients in leaflet format at the onset of the study [25]. 

The EI has been previously described in detail [26], and comprised three to five aerobic exercise sessions per week increasing in number over time, with two sessions led by an exercise specialist and an additional one to three unsupervised sessions prescribed. Supervised exercise sessions (twice a week) were conducted on treadmills, cycle ergometers, and elliptical trainers. Each session consisted of a 5–7 min warm-up session, an aerobic duration (21–42 min, which increased throughout the 12 weeks of the EI), and a 5–7 min cool-down session. The intensity of exercise was individualised, determined by each participant’s heart rate reserve, and progressively increased (40–75% heart rate reserve), along with the aerobic duration, throughout the EI. For unsupervised exercise sessions, participants were encouraged (by text message weekly) to replicate the intensity, type, and duration of the supervised exercise sessions that were prescribed for each respective week as much as possible. Unsupervised sessions began at 1 time per week for weeks 1–3 but were prescribed to increase to twice a week for weeks 4–7 and 3 times per week for weeks 8–12 of the EI. Adherence to unsupervised sessions were monitored through the completion of exercise diaries. 

Following completion of the exercise and dietary interventions, all participants were reassessed at week 13 (T1).

### 2.4. Dietary and Cardiorespiratory Assessment

At T0, all participants were administered an NAFLD-targeted and validated 48-item food frequency questionnaire [27] in an interview with a trained research nutritionist that lasted approximately 20 min. Participants were then advised on how to complete a detailed written four-day diet diary on two weekdays and two weekend days, for return by mail. The four-day diet diary template had three columns for recording meal (i.e., breakfast, lunch, evening meal, snack), time of meal, and the weights and sizes of foods. Participants were encouraged to give as much detail as practical on portion size using information from package information and household measures and to weigh foods whenever possible. 

At T1, the same research nutritionist re-interviewed participants and participants were again asked to complete a detailed four-day diary. Nutrient intakes were analysed using the online, myfood24™ dietary assessment tool, which incorporates both the UK food composition dataset of ~3300 items and an additional >40,000 generic and branded items commonly found in UK and Irish supermarkets [28]. Diaries were inputted as four 24 h periods, and mean intakes per day were calculated for each nutrient for each participant. Overall diet quality was systematically assessed from the food frequency questionnaire data using a scoring system where each question was scored on a scale of 0 to 5 with optimal intakes being allocated a score of 5.

#### 2.4.1. Cardiorespiratory Fitness Assessment

As described previously [26], prior to fitness testing, all participants were screened for cardiovascular risk and physical activity readiness. Cardiorespiratory fitness was assessed using the Modified Bruce submaximal cardiopulmonary exercise test protocol [29] on an electrically driven treadmill to give estimates of maximal oxygen consumption (VO_2max_). Gas exchange variables were collected using a portable indirect calorimeter, and heart rate was measured continuously using a chest strap. Participants performed the exercise protocol until 85% of their age-predicted maximal HR was attained, or they reached volitional fatigue. VO_2max_ was estimated using the American College of Sports Medicine metabolic equation [29].

#### 2.4.2. Dietary and Cardiorespiratory Assessments

Dietary intakes of all participants were assessed at T0 and T1, both by the four-day diet diaries returned by mail, and by a validated [27], NAFLD-targeted food frequency questionnaire administered via a 20 min interview by a trained nutritionist. In EI participants, cardiorespiratory fitness (estimated maximal oxygen consumption, VO_2max_) was assessed using a submaximal cardiopulmonary exercise test protocol on a treadmill using a portable indirect calorimeter and heart rate monitor as detailed previously [26]. Both dietary and cardiorespiratory assessments are described further in Supplementary Methods.

### 2.5. Histological Analysis and Isolation of MAIT Cell Populations

#### 2.5.1. Transient Elastography

Vibration-controlled transient elastography (VCTE) was performed using a FibroScan^®^ touch 502 (Echosens, Paris, France) with M and XL probes to non-invasively estimate hepatic fibrosis (liver stiffness score) and steatosis (controlled attenuation parameter, CAP) at T0 and T1.

#### 2.5.2. Liver Histology

All participants had liver biopsies at T0. Patients with NASH in the DI and EI groups had repeat biopsies at T1. All liver biopsy specimens were reviewed and scored by a single, blinded histopathologist. Hepatic steatosis was scored based on the proportion of hepatocytes affected and subsequently classed into four grades (0–3). The severity of liver injury was assessed and scored using the NASH Clinical Research Network criteria [30]. The NAFLD activity score (NAS) was graded between 0 and 8, and hepatic fibrosis was staged between 0 and 4. 

#### 2.5.3. Tissue Collection for Analysis of Circulating and Intra-Hepatic MAIT Cell Populations

Whole blood and paired liver biopsies were obtained for phenotyping of circulating and intrahepatic immune cell populations by flow cytometry at T0 and T1. Liver biopsies were obtained by an ultrasound-guided percutaneous procedure [31]. Whole blood samples were taken on the same day as the liver biopsy. Liver and whole blood specimens were processed immediately and underwent analysis within 3 h of sampling.

#### 2.5.4. Preparation of Liver Immune Cells

Liver biopsy tissue was digested enzymatically in Hank’s buffered saline solution (HBSS, BD Biosciences; Wokingham, UK) containing 2% foetal bovine serum, 2% bovine serum albumin, and 125 U/mL type IV collagenase (all, Sigma-Aldrich Ltd., Wicklow, Ireland) as previously described [32]. Tissue was incubated for 25 min on a shaking incubator, at maximum agitation, at 37 °C at 150 revolutions per minute (RPM) before being passed through a 70 μm polypropylene filter (BD Biosciences; Wokingham, UK). Cells were washed twice, initially with HBSS and subsequently with phosphate-buffered saline (PBS, BD Biosciences; Wokingham, UK), and then centrifuged at 1300 RPM for 3 min. The pellet was re-suspended in PBS prior to staining.

#### 2.5.5. Immunophenotyping by Flow Cytometry

Whole blood and intrahepatic mononuclear single-cell suspensions were stained with the following: anti-CD45-PerCP (clone 2D1), anti-CD3-APC-H7 (clone SK7), anti-CD8-450 (clone RPA-T8), anti-CD161-APC (clone DX12), anti-CD69-PeVio770 (clone FN50), and anti-CD95-PE (clone DX2), all from BD Biosciences (Wokingham, UK), as well as anti-Vα7.2-FITC (clone REA 179; Miltenyi Biotech LTD, Bisley, UK) and Zombie Aqua Viability Dye (BioLegend UK Ltd., London, UK). MAIT cells were identified as CD3^+^ Vα7.2^Hi^ CD161^Hi^ populations. Data were acquired on a clinical grade FACSCanto^TM^ II flow cytometer using the FACSDiva^TM^ software, and then analysed with FlowJo v10.8 software (all, BD Biosciences, Wokingham, UK).

### 2.6. Statistical Analysis

All statistical analyses were conducted using GraphPad Prism v9.0 (GraphPad Software Inc., La Jolla, CA, USA). As there were no data on the response of intrahepatic MAIT cells to lifestyle interventions on which to base a formal power calculation, in line with the appropriate CONSORT extension [33], a primary aim of this pilot and feasibility study was to generate the data to inform the design of a definitive trial. Obtaining paired liver biopsies is not trivial with ethical considerations around the invasive nature of this procedure. Therefore, the decision to recruit a total of 50 participants was a pragmatic one, in line with the single immunophenotyping study done on liver biopsies from patients with NAFLD that had an *n* = 15 [23]. Data were tested for normality using the Shapiro–Wilk test and continuous variables presented as either mean ± standard deviation or as median (interquartile range). Baseline between-group differences were assessed using an ordinary one-way ANOVA or Kruskal–Wallis test as appropriate. Categorical variables are presented as number (percentage) and distributions were compared using the Chi-square test. The paired *t*-test or Wilcoxon matched pair signed-rank test were used to assess differences between repeated measurements within each intervention group. Statistical significance for all tests was set at *p* ≤ 0.05.

## 3. Results

### 3.1. Baseline Characteristics

Of the 50 participants enrolled in the study, 45 completed the study (CG, *n* = 14; DI, *n* = 15; EI, *n* = 16; Figure 1). At baseline, these participants were well matched with no differences in clinical and histological characteristics (Table 1). 

Participants presented with typical NAFLD comorbidities, as 77% of the overall cohort were obese (body mass index, BMI ≥ 30 kg/m^2^) and the remaining 23% were overweight (BMI 25–29.9 kg/m^2^). Many had either impaired glucose tolerance or type 2 diabetes (64%), as well as hypertension (53%) and metabolic syndrome (51%). 

The majority (84%) met the histological criteria for NASH; 64% had advanced fibrosis with Brunt stage ≥ 2, and 41% had a NAFLD activity score (NAS) ≥ 5. Six participants (3 in DI, 3 in EI) were assessed as NAFL (steatosis only) on biopsy at T0 and did not have repeat biopsies.

### 3.2. Improvements in Clinical Parameters with Diet and Exercise

Although both the DI and EI elicited significant reductions between T0 and T1 in body weight, BMI, waist circumference, and HbA1c, greater loses were observed in the DI group (Table 2). While the DI elicited a mean weight loss of 7 kg (*p* < 0.0001), the EI participants lost a mean of 2 kg (*p* = 0.0005). Similarly, mean reductions in BMI (DI: −1.9 kg/m^2^, *p* = 0.0002; EI: −1.1 kg/m^2^, *p* < 0.0001) and waist circumference (DI: −11.6 cm, *p* < 0.0001; EI: −4.0 cm, *p* < 0.0001), along with median changes in HbA1c (DI: −3.0 mmol/mol, *p* = 0.0027; EI: −1.5, *p* = 0.0298), were more pronounced in the DI participants in comparison to EI. Similarly, modest weight loss has been reported among NALFD patients with aerobic exercise in comparison to nutritional intervention [12,34]. In contrast, no changes were observed in body weight, BMI, weight circumference, and HbA1c in participants in the CG between T0 and T1 (Table 2).

While VCTE scores for steatosis and liver stiffness improved significantly in both DI and EI participants, improvements were greater in EI participants. A 12.5% reduction in the mean controlled attenuation parameter (CAP) score (T0: 330.4 ± 47.4 versus T1: 289.2 ± 43.2 dB/m; *p* = 0.0036) was observed in DI, and a 13.8% reduction (T0: 334 ± 43.4 versus T1: 288.3 ± 73.9 dB/m; *p* = 0.0033) was observed in EI participants. While DI participants showed a 20.8% reduction in the median liver stiffness score (T0: 12.0 (4.5) versus T1: 9.5 (4.8) kPa; *p* = 0.0154), EI participants had a reduction of 27.6% (T0: 12.3 (7.1) versus T1: 8.9 (5.4) kPa; *p* = 0.0038). No changes in CAP or liver stiffness scores were observed in CG participants (Table 2). 

Liver enzymes were more variably decreased across groups. Median ALT values (T0: 45.0 (70.0) versus T1: 32.0 (32.0) IU/L; *p* = 0.0054) and GGT (T0: 61.0 (61.0) versus T1: 38.0 (45.0) IU/L; *p* < 0.0001) were significantly decreased in DI participants; changes in median AST levels were not significant (*p* = 0.0777). Only GGT levels decreased significantly in EI participants (T0: 57.5 (91.5) versus T1: 44.0 (69.8); *p* = 0.0287). Although unexpected decreases in ALT and AST levels in the CG were observed between T0 and T1 (Table 2), this was driven by a single participant who had very high ALT and AST levels at the T0 assessment that decreased by >75% by T1 (Appendix A) as a result of the patient’s self-motivation to follow the standard of care guidance to lose weight. There were no significant differences in the FIB4 index in the intervention groups; however, a significant reduction was seen in the control group (T0: 1.57 (0.72) versus T1: 1.30 (0.61), *p* = 0.0284. Significant reductions in the FAST score were observed in the control (T0: 0.68 ± 0.22 versus T1: 0.57 ± 0.23, *p* = 0.0399) and DI groups (T0: 0.554 ± 0.23 versus T1: 0.38 ± 0.25, *p* = 0.0042) but not in the EI group. 

Adherence to both the DI (87%) and EI (93%) was high. Significant reductions in energy intakes (*p* = 0.0045) and improvements in dietary quality (*p* < 0.000001) were observed post intervention in DI, but not EI, participants (Appendix A). Importantly, reductions in energy intakes in the DI group were driven by significant decreases in total intakes (g/day) of saturated fat (*p* = 0.0260), carbohydrates (*p* = 0.0001), and total sugars (*p* = 0.0024) (Appendix A). In contrast, there were no changes in dietary quality or energy or macronutrient intakes in the CG or EI participants (Appendix A). As previously reported [26], EI participants significantly improved their cardiorespiratory fitness in response to intervention, as measured by estimated VO_2max_ during a submaximal cardiopulmonary exercise test. There were no changes to VO_2max_ observed in the CG, and DI participants did not undergo cardiorespiratory fitness testing. 

### 3.3. Differential Improvement in Histological Outcomes between Diet and Exercise

At baseline, 12/15 (80%) of DI and 13/16 (81%) of EI participants had NASH. Participants with simple steatosis at T0 did not have repeat biopsies at T1 and therefore were excluded from paired histological analyses. In addition, a participant who started methotrexate during the EI intervention was also excluded, resulting in *n* = 12 paired biopsies for examination of histological changes post-intervention for both the DI and EI groups (Figure 1).

For each component of NAS and fibrosis grading, the proportion of patients who were stable, improved, or worsened was examined (Figure 2), in addition to individual participant changes (Appendix A). EI participants showed significant improvements in Brunt fibrosis stage (58.3% improved, *p* = 0.0352; Figure 2A and Appendix A) and hepatocyte ballooning (66.7% improved, *p* = 0.0195; Figure 2B and Appendix A). Further, 16.7% of EI participants showed an increase in steatosis grade. In contrast, significant improvements in steatosis (66.7% improved, *p* = 0.0039; Figure 2C and Appendix A) and NAS (66.7% improved, *p* = 0.0098; Figure 2D and Appendix A) were observed in the DI group. This differential effect in steatosis grade was associated with changes in dietary quality, particularly correlating with a reduction in sugar intake (r = 0.7534, *p* = 0.0093). Appendix A highlight the differential effects of the interventions on dietary quality and macronutrient intakes. There were no significant changes for lobular inflammation after 12 weeks of either DI (Figure 2E and Appendix A) or EI (Figure 2F and Appendix A).

### 3.4. Alterations in Circulating and Intrahepatic MAIT Cell Populations

Circulating MAIT cells were defined as the CD3^+^ Vα7.2^Hi^ CD161^Hi^ population of CD45^+^ lymphocytes, with the doublet exclusion part of the gating strategy (Figure 3A). Representative histograms of distribution of the CD69 and CD95 surface marker expression in circulating MAIT cells are shown in Appendix A. Approximately 1% of the circulating CD3^+^ lymphogates were Vα7.2^Hi^ CD161^Hi^ (Figure 3B). There was a trend towards a reduced percentage of circulating MAIT cells with advanced NAFLD fibrosis (Brunt stage ≤ 2: 1.260 (1.475) versus Brunt > 2: 0.960 (0.505) that did not achieve statistical significance, *p* = 0.0829. Expression (median fluorescence intensity, MFI) of the acute activation marker CD69 was in general quite low in circulating MAIT cells, and DI participants showed decreased CD69 expression (T0: 104 (134) versus T1 27 (114); *p* = 0.0353), but with a notable variance in expression. In contrast, expression of CD95, the Fas death receptor, was at least 10-fold higher than CD69, and significantly increased 1.65-fold in response to EI (T0: 1549 (888) versus T1: 2563 (1371) MFI, *p* = 0.0043; Figure 3C).

Intrahepatic MAIT cells were similarly gated (Figure 4A) and were notably more abundant in the liver, representing ~5–7% of the CD3^+^ population (Figure 4B), than in the circulation. Although CD69 was much more robustly expressed in intrahepatic MAIT cells than in circulating MAIT cells (Figure 4C), there were no differences observed at T1 for either intervention (Figure 4C). However, the percentage of intrahepatic MAIT cells significantly decreased in response to EI (T0: 11.1 (14.4) versus T1: 5.3 (9.3)%, *p* = 0.0029; Figure 4B). Moreover, this occurred alongside a significant increase in detected CD95 expression in response to EI (T0: 2724 (862) versus T1: 3117 (1622) MFI, *p* = 0.0269; Figure 4D). 

## 4. Discussion

This is the first study to investigate circulating and intrahepatic MAIT cell populations in the context of metabolic and histological changes in patients with NAFLD who completed either a 12-week dietary or aerobic exercise intervention. While both DI and EI participants demonstrated improved clinical and transient elastography parameters of NAFLD, weight reductions were more pronounced in DI participants, who additionally demonstrated significant improvements in histological steatosis grade and reduced expression of activation markers on circulating MAIT cells. In contrast, significant improvements in fibrosis stage and hepatocyte ballooning were observed in EI participants, associated with increased apoptotic marker expression and a significant reduction in intrahepatic MAIT cells numbers.

Circulating MAIT cells are markedly decreased in adults with obesity and/or type 2 diabetes, as well as children with obesity [18,19,35]. Moreover, circulating and adipose residing MAIT cells are activated with a pro-inflammatory (Th17) phenotype, producing less IFNγ, but higher levels of TNFα and IL-17. Multiple studies have shown positive shifts after bariatric surgery-induced weight loss from proinflammatory to anti-inflammatory phenotypes in lymphocyte populations, including MAIT cells, and cytokine secretion [36]. These shifts include decreased expression of the early activation marker CD69 [37,38], and increases in circulating MAIT cell numbers post bariatric surgery [18,35]. However, the data examining the expression of inflammatory markers indicate that not all markers revert to levels observed in healthy controls [36]. In a study of bariatric surgery patients compared to healthy controls, IL-17 levels remained high despite serum levels of IL-2 and granzyme B reducing in line with controls at 6- and 12-months post-surgery [18]. However, no studies to date have examined changes in MAIT cell populations in response to weight loss from dietary or exercise interventions or have investigated for these changes in patients with NAFLD.

In our study, low levels of circulating MAIT cells as percent of all CD3^+^ lymphocytes were observed both pre- and post-intervention, similar to reports of low numbers of MAIT cells in participants with obesity [18,19]. Although we did not observe changes in the total percentage of circulating MAIT cells in response to either DI or EI, expression of the early activation marker CD69 was reduced in circulating MAIT cells from participants who completed the 12-week DI. This is consistent with earlier work suggesting weight loss reduces lymphocyte activation [37,38]. While EI participants did decrease their bodyweight, the reductions were not as pronounced as those in the DI group, and notably, no alterations of CD69 expression in MAIT cells were seen in response to EI. It may be that a greater and/or longer sustained period of weight loss may be needed to promote significant increases in the percentage of circulating MAIT cells. 

We observed increased expression of the apoptotic marker CD95, the Fas death receptor, in both circulating and intrahepatic MAIT cells after EI. While stimulation of MAIT cells promotes expression of CD69, excessive chronic activation promotes CD95 expression on surface membranes and subsequent binding of the Fas ligand activates death caspases and cell apoptosis [39]. Interestingly, the increased expression of CD95 in intrahepatic MAIT cells after EI was observed in tandem with an apparent reduction in the percentage of intrahepatic MAIT cells, in addition to improvements in liver fibrosis and hepatocyte ballooning. In contrast, while the DI participants lost more weight and showed significant improvements in steatosis score, no improvements in fibrosis were observed. Animal studies have shown that MAIT cell-enriched mice exhibited accelerated hepatic fibrogenesis in comparison to MAIT cell-deficient mice [21]. These data might suggest that targeting MAIT cells may constitute an attractive antifibrogenic strategy in chronic liver disease. However, MAIT cells have also demonstrated anti-bacterial and anti-viral activity; therefore, it is likely that intrahepatic MAIT cells may play a protective role against gut-derived pathogens. 

Only a limited number of studies have examined circulating MAIT cells in response to exercise. Circulating MAIT cells have been shown to increase in response to both an acute graded exercise test to volitional exhaustion [40], and a 40 min submaximal test [41], although MAIT cell numbers rapidly return to baseline within an hour [41]. Exercise training in breast cancer survivors improved responsiveness of MAIT cells observed after 45 min of intermittent cycling, although not to the level observed in healthy older women [42]. Both acute and chronic exercise programs have been shown to influence the immune system [43,44]. Although the heavy exercise workloads of competitive athletic training have been associated with inflammatory immune effects, regular (moderate) exercise training has an anti-inflammatory effect that is believed to have a summative effect over time in cardiometabolic disease and cancer prevention [45]. Obtaining paired liver biopsies is not trivial and likely explains why these are the first data examining intrahepatic MAIT cells in response to lifestyle intervention in NAFLD.

In our cohort, improvements in lobular inflammation and Brunt fibrosis stage were observed in EI participants despite lower mean weight loss, whereas DI participants, with a mean loss of 7 kg, exhibited reductions in steatosis grade alone. This differs from a previous trial in NAFLD patients that correlated histological improvements with the degree of weight loss [14]. In this study, follow-up liver biopsies were performed after 6 months. The improvements observed in the EI participants may reflect earlier immunological changes in association with increased cardiorespiratory fitness, while it is possible that the metabolic improvements caused by weight loss may promote more gradual improvements in NASH and NAFLD fibrosis.

Our study had some limitations. It had a relatively small sample size, with *n* = 45 completing the study, and participants were allocated to the intervention groups based on preference and were not randomized. A combined exercise and nutritional intervention arm in addition to a healthy control group would have further strengthened the study. However, as liver biopsies were performed in all study participants, for ethical reasons, healthy controls were not recruited. Similarly, paired liver biopsies were not performed in the control group. Although changes in weight, BMI, and waist circumference were recorded, changes in body composition, percentage body fat, and muscle mass using bioimpedance or dual energy x-ray absorptiometry were not measured. Participants were given non-personalised dietary recommendations and adherence was assessed using four-day food diaries. While dietary assessments were made before and after the EI, physical activity levels were not assessed in the DI group. The novel immune experiments focused primarily on characterising circulating and intrahepatic MAIT cells and did not assess for concomitant changes in other immune cell populations that may also contribute to the histological changes observed. Lastly, experiments focused on MAIT cell percentages and cell surface marker expression, and no measurements of MAIT cell metabolism were made.

Nevertheless, to our knowledge, this study is the first to examine intrahepatic MAIT cells in response to lifestyle intervention. MAIT knockout mice are protected from fibrosis in models of chronic liver injury, and it is tempting to speculate that our observed increase in CD95 and decreased frequency of intrahepatic MAIT cells may explain the histological improvements in fibrosis and hepatocyte ballooning that we observed in the EI participants. However, whether intrahepatic MAIT cells are protectors or protagonists in the context of NAFLD pathogenesis is a question that has not yet been resolved, and the results of this pilot study will need be extended to a larger population. Although MAIT cells have been demonstrated to promote inflammation in both adipose and the gut of high fat diet-fed mice [46], MAIT knock out mice experience more severe hepatic steatosis and inflammation when fed a methionine choline-deficient diet, suggesting MAIT cells could play a protective role in NAFLD-related inflammation [22]. Recent reports show the existence of a tissue repair gene signature in liver-derived MAIT cells, which can be induced by TCR activation [24]. More studies examining the function of MAIT cells in patients with NAFLD in response to diet and exercise alone and in combination are needed to better understand the role of MAIT cells in disease pathogenesis in patients with NASH and other chronic liver diseases [47]. 

## 5. Conclusions

The results of this study demonstrate that, in contrast to a 12-week dietary weight loss program, a 12-week aerobic exercise program resulted in improvements in fibrosis and ballooning stage in patients with NAFLD. These improvements occurred alongside increased apoptotic marker expression and a significant reduction in intrahepatic MAIT cell numbers in EI participants. In contrast, DI participants showed decreased circulating expression of the activation marker CD69 and improvements in histological steatosis grade post-intervention. These data demonstrate independent benefits from dietary and exercise intervention and suggest the involvement of intrahepatic MAIT cells in the observed improvements in the histological features of NAFLD. Future research investigating the role of lifestyle in regulating MAIT cells in health and chronic liver disease is warranted.

## Figures and Tables

**Figure 1 nutrients-14-02198-f001:**
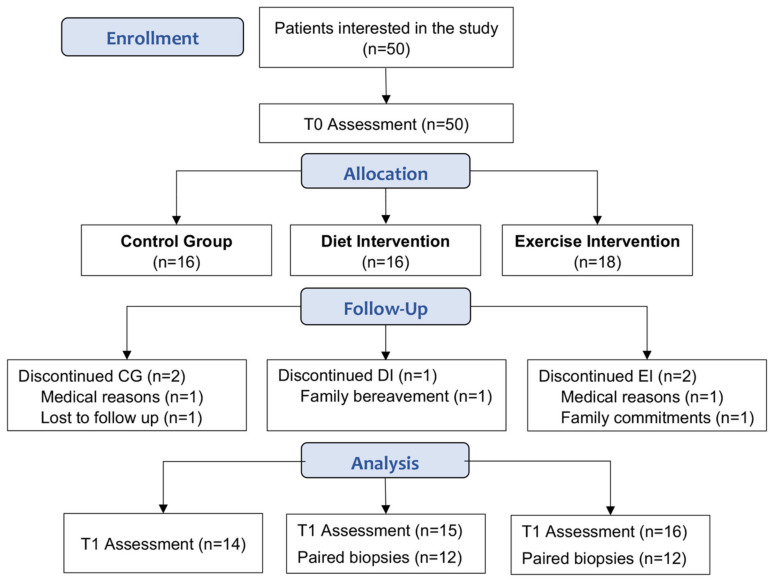
Participant recruitment and attrition. T0: baseline assessment, T1: post intervention (week 13).

**Figure 2 nutrients-14-02198-f002:**
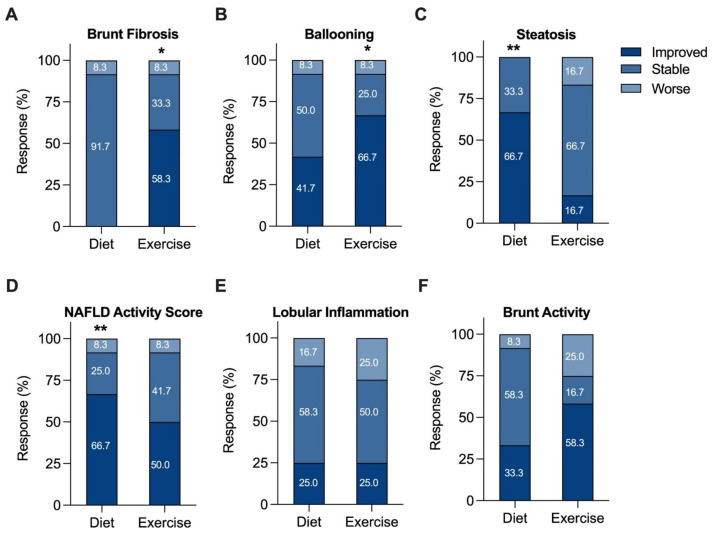
Changes in histologic categories (improved, stable, or worse) after a 12-week diet or exercise intervention. Any increase or decrease in score was considered a change in category. (**A**) Fibrosis, (**B**) ballooning, (**C**) steatosis, (**D**) NAFLD activity, (**E**) lobular inflammation, and (**F**) Brunt activity scores. The percent of patients in each category are shown numerically within each category of the bar graph. * *p* <0.05, ** *p* <0.01 from baseline, Wilcoxon matched pairs signed-rank test.

**Figure 3 nutrients-14-02198-f003:**
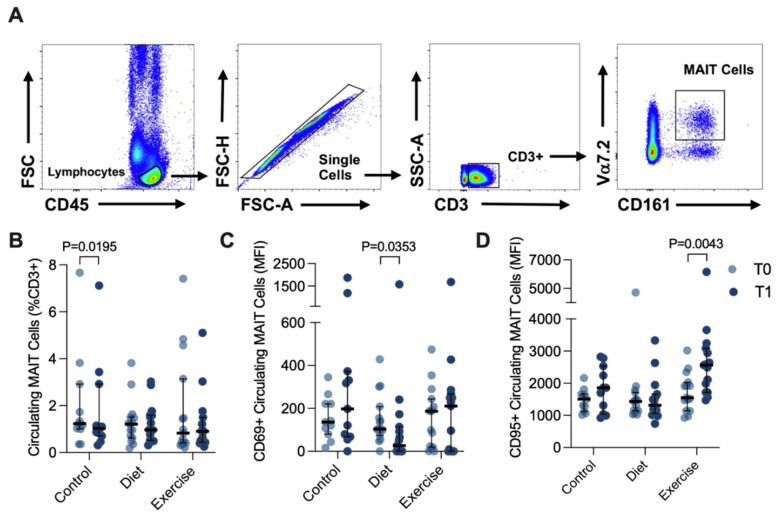
Circulating MAIT cells before (T0) and after (T1) diet or exercise intervention. (**A**) Representative flow cytometry dot plots showing gating strategy for defining circulating CD3^+^ Vα7.2^hi^ CD161^hi^ MAIT cells. (**B**) Circulating MAIT cells in control participants (*n* = 11, as 3 participant samples were lost during processing), and participants completing either 12 weeks of dietary (*n* = 15) or exercise (*n* = 15) intervention. (**C**) CD69+ and (**D**) CD95+ circulating MAIT cells before and after intervention. Data were analysed via Wilcoxon matched pairs signed-rank test, and individual values and median ± interquartile range are depicted. MFI: median fluorescence intensity. MAIT: Mucosal-associated invariant T.

**Figure 4 nutrients-14-02198-f004:**
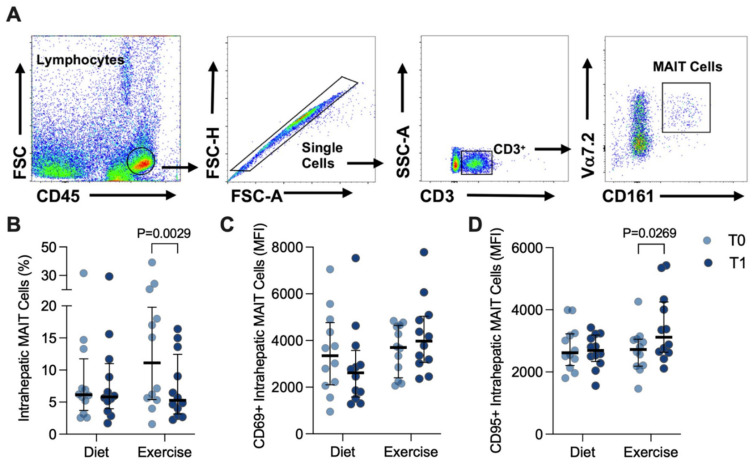
Intrahepatic MAIT cells before (T0) and after (T1) diet or exercise intervention. (**A**) Representative flow cytometry dot plots showing gating strategy for defining intrahepatic CD3^+^ Vα7.2^hi^ CD161^hi^ MAIT cells. (**B**) Intrahepatic MAIT cells in participants completing either 12 weeks of diet (*n* = 12) or exercise (*n* = 12) intervention. (**C**) CD69+ and (**D**) CD95+ circulating MAIT cells before and after intervention. Data were analysed via Wilcoxon matched pairs signed-rank test and individual values and median ± interquartile. MAIT: Mucosal-associated invariant T.

**Table 1 nutrients-14-02198-t001:** Baseline clinical and laboratory characteristics.

Parameters	Control (*n* = 14)	Diet (*n* = 15)	Exercise (*n* = 16)	*p* Value
Age (y)	55 (20)	58 (14)	61 (15)	0.3581 ^a^
Gender, n (%)				
Female	7 (50.0%)	8 (53.3%)	12 (75%)	0.5331 ^b^
Male	7 (50.0%)	7 (46.7%)	4 (25%)	
Body weight (kg)	102.8 ± 33.3	97.3 ± 22.3	95.6 ± 20.0	0.7276 ^c^
BMI (kg/m^2^)	35.9 ± 7.1	33.9 ± 5.3	36.1 ± 6.9	0.5996 ^c^
Hypertension n (%)	7 (50.0%)	8 (53.3%)	9 (56.3%)	0.9431 ^b^
Diabetes/IGT n (%)	10 (71.4%)	8 (53.3%)	11 (68.8%)	0.5391 ^b^
Metabolic Syndrome	7 (50.0%)	7 (46.7%)	9 (56.3%)	0.8630 ^b^
HbA1c (mmol/mol)	46 (19)	40 (8)	47 (24)	0.2126 ^a^
Urate (μmol/L)	365 ± 99	351 ± 95	324 ± 68	0.4362 ^c^
Vitamin D (nmol/L)	33 (20)	52 (26)	56 (57)	0.1632 ^a^
Triglycerides (mmol/L)	1.8 (1.4)	1.9 (1.4)	1.4 (1.4)	0.4797 ^a^
LDL-C (mmol/L)	2.1 ± 0.9	2.4 ± 1.0	2.3 ± 0.8	0.7501 ^c^
HDL-C (mmol/L)	1.3 ± 0.3	1.2 ± 0.3	1.4 ± 0.5	0.1055 ^c^
ALT (IU/L)	66 (37)	45 (70)	47 (26)	0.2763 ^a^
AST (IU/L)	56 (30)	33 (47)	33 (14)	0.1371 ^a^
GGT (IU/L)	113 (148)	61 (61)	58 (92)	0.1420 ^a^
Brunt Fibrosis				
Stage 0/1	4 (28.6%)	7 (46.7%)	5 (31.3%)	N/A ^d^
Stage 2	2 (14.2%)	2 (13.3%)	4 (25.0%)	
Stage 3	4 (28.6%)	4 (26.7%)	5 (31.3%)	
Stage 4	4 (28.6%)	2 (13.3%)	2 (12.5%)	
Histological scoring				
(NAS < 5)	5 (35.7%)	8 (56.3%)	10 (62.5%)	0.6355 ^c^
(NAS ≥ 5)	9 (64.3%)	7 (43.7%)	6 (37.5%)	

Data were tested for normality using Shapiro–Wilk test and continuous variables presented as either mean ± standard deviation or as median (interquartile range) as appropriate. Categorical variables are presented as number (percentage). ALT, alanine aminotransferase; AST, aspartate aminotransferase; BMI, body mass index; GGT, gamma glutamyl transferase; HbA1c, haemoglobin A1c; IGT, impaired glucose tolerance; NAS, NAFLD activity score. ^a^ Kruskal–Wallis test. ^b^ Chi-square test. ^c^ One-way ANOVA. ^d^ Conditions for Chi-square not met.

**Table 2 nutrients-14-02198-t002:** Changes in clinical parameters from baseline (T0) to week 13 (T1).

	Control (*n* = 14)			Diet (*n* = 15)			Exercise (*n* = 16)		
Parameters	T0	T1	*p* Value	T0	T1	*p* Value	T0	T1	*p* Value
Weight (kg)	102.8 ± 33.3	102.8 ± 31.6	0.9899	97.0 ± 22.3	90.0 ± 19.9	<0.0001	95.6 ± 20.0	93.6 ± 19.8	0.0005
BMI (kg/m^2^)	35.9 ± 7.1	36.4± 7.7	0.1422	33.9 ± 5.3	32.0 ± 5.3	0.0002	36.1 ± 7.0	35.0 ± 6.7	<0.0001
WC ^a^ (cm)	105.4 ± 14.0	105.2 ± 14.1	0.8911	119.9 ± 16.0	108.3 ± 14.9	<0.0001	111.2 ± 15.3	107.2 ±15.7	<0.0001
HbA1c (mmol/mol)	46.0 (18.8)	43.0 (22.8)	0.9629	40.0 (8.0)	37.0 (6.0)	0.0027	46.5 (23.8)	45.0 (23)	0.0298
CAP (dB/m)	325.6 ± 64.7	341.0 ± 56.8	0.2149	330.4 ± 47.4	289.2 ± 43.2	0.0036	334.4 ± 43.4	288.3 ± 73.9	0.0033
Liver Stiffness (kPa)	17.1 (9.7)	13.7 (12.1)	0.2166	12.0 (4.5)	9.5 (4.8)	0.0154	12.3 (7.1)	8.9 (5.4)	0.0038
FAST score	0.68 ± 0.22	0.57 ± 0.23	0.0399	0.54 ± 0.23	0.38 ± 0.25	0.0042	0.51 ± 0.21	0.43 ± 0.18	0.0838
FIB-4 score	1.57 (0.72)	1.30 (0.61)	0.0284	1.34 (0.89)	1.44 (0.70)	0.2314	1.32 (0.67)	1.27 (0.81)	0.137
ALT (IU/L)	66.0 (37.3)	47.0 (46.8)	0.0342	45.0 (70.0)	32.0 (32.0)	0.0054	46.5 (25.5)	40.5 (20.5)	0.0856
AST (IU/L)	56.0 (29.8)	34.5 (21.3)	0.0289	33.0 (47.0)	30.0 (28.0)	0.0777	33.0 (14.0)	33.0 (11.0)	0.1682
GGT (IU/L)	113.0 (147.8)	84.0 (103.5)	0.5114	61.0 (61.0)	38.0 (45.0)	<0.0001	57.5 (91.5)	44.0 (69.8)	0.0287

Data were tested for normality using Shapiro–Wilk test and presented as either mean ± standard deviation or as median (interquartile range), with differences assessed by paired *t* test or Wilcoxon matched pair signed-rank test as appropriate. ALT, alanine aminotransferase; AST, aspartate aminotransferase; BMI, body mass index; CAP, controlled attenuation parameter; GGT, gamma glutamyl transferase; HbA1c, haemoglobin A1c; FAST score, Fibroscan-AST score, FIB-4, Fibrosis-4 index, WC, waist circumference. ^a^ WC control group *n* = 8 only.

## Data Availability

Data are contained within the article and Appendix A.

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
