# Peer review of "Differential Effects of Dietary versus Exercise Intervention on Intrahepatic MAIT Cells and Histological Features of NAFLD"

_nutrients, 2022, doi:10.3390/nu14112198_

Round 1

Reviewer 1 Report

In their comprehensive study Naimimohasses et al. analyzed the effects of either 12 weeks of dietary intervention (DI) or aerobic exercise intervention (EI) on circulating and intrahepatic mucosal associated invariant T (MAIT) cell populations in 45 patients with NAFLD by using multicolour flow cytometry immunophenotyping.

In detail, MAIT cell populations, liver histology and clinical parameters  were assessed at baseline (T0) and following completion (T1) of the dietary intervention or aerobic exercise intervention.

Here, the study authors observed that the percentage of intrahepatic MAIT cells significantly decreased after exercise intervention and was associated with significant improvements in fibrosis stage and hepatocyte ballooning. These changes were accompanied by an increased expression of the apoptotic marker CD95 of MAIT cells, both in circulating and intrahepatic MAIT cells.

When focusing on the effects of dietary interventions, the authors observed a decreased expression of the activation marker CD69 on circulating MAIT cells along with improvements in histological steatosis grade post-intervention.

Based on their observations, the authors conclude that intrahepatic MAIT cells may be independelty associated with histological improvements in NAFLD patients undergoing certain interventions such as aerobic exervise or dietary changes.

While the manuscript is of interest and examines an immune cell population for which less is known in the context of NAFLD pathogenesis until today, some aspects should be revised before publication is considered:

Introduction:

The authors stated that in patients with NAFLD, the levels of circulating MAIT cells are decreased in comparison to

healthy controls and refer to specific references.

In the underlying study setting the majority of patients  met the histological criteria for NASH ( 64%) had advanced fibrosis  (41%) indicating progressive disease stages. Do the levels of circulating cells also vary depending on NAFLD disease severity?

As the severity of the disease is of particular prognostic relevance (as also described by the authors at the beginning of the introduction), this information would be relevant and should be added.

Methods:

Why was a healthy control group omitted ?
Besides the CG, DI and EI group in the underlying study, an additional group of healthy individulas without NAFLD would allow further comparisons

and would give additional information about the normal MAIT cell distribution in the underlying study setting.

The authors should justify the omission of this group.

How was it ensured and verified that the study participants adhered to the dietary recommendations ?
Were all diet diaries regularly checked, except To and T1. Was regular screening also carried out during the study weeks between T0 and T1 ?
Please elaborate.

Results:

How do the authors explain such significant differences in weight loss between the two groups?

( D participants: 7 kg versus EI participants: 2 kg)

Was the exercise training, especially the unsupervised sessions, not done sufficiently ?

Table 2:

It would be informative if the values of the Fib4 score, the NAFLD score and the FAST score were implemented in table two and compared between the different time points, if this is possible.

Figure 2.3:

How do the study authors explain a worsening of steatosis in about 17 percent of patients in the exercise group ?

Discussion:

The  authors conclude that the results of their study demonstrate that, in contrast to a 12-week dietary weight loss program, a 12-week aerobic exercise program resulted in improvements in fibrosis and ballooning stage in patients with NAFLD.

The histological changes in the exercise group were seen although the weight loss was significantly lower compared to the dietary intervention group.
Previous studies in this context  in NAFLD patients were able to show that a decrease in fibrosis commonly correlates closely with the extent of body weight loss. Why does this not seem to be the case here ?
The authors should describe and address this aspect in more detail.

General considerations that could be addressed in more detail within the discussion:

-Are there pharmacological options targeting MAIT cells t( e.g. neutralizing antibodies)

-The liver is also an immunological organ that plays a central role in the recognition and defence of gut derived DAMPs and PAMPs and can induce immunological responses against circulating pathogens within the portal circulation.Can a decrease in intrahepatic MAIT cell number also be associated with potential negative consequences?

Minor comments:

The text should be revised regarding the use of Abbreviation:

e.g. within the introduction, all abbreviations should be spelled out if they are established in the text for the first time (e.g. IFN, TNFalpha and IL-17 et cetera)

Author Response

We would like to thank the reviewers for their care and comprehensive review of our manuscript. We have taken their useful feedback onboard  and strived to address each point as described in the file attached

Reviewer 2 Report

The authors present an interesting aspect of the action of physical activity and physical exercise.

The results encourage a lifestyle intervention regarding diet and physical activity and I completely agree on this aspect, it would have been very interesting to evaluate the combination of both that should be.

There are some points to clarify:

line 131 how can you say ad libitum? what was the recommendation?

looking at the caloric intakes there are enormous variations, especially in the control group, this should at least be emphasized, probably the subjects were divided into different groups.

Was the split random? Double-blind? How come those in the control group started from a higher caloric intake, especially when compared to those who exercise?

How were the changes in the diet assigned after analyzing the diary?

Although not statistically significant, how do you justify an average decrease of 200kcal in the control group?

Important limit of the work not to have used at least one bioimpedance, even better it would have been in iDXA, varying the loss may not be significant, especially when practicing physical activity, obviously, it cannot be done a posteriori, but should be emphasized in the limitations of the manuscript.

As underlined in the conclusions, the two paths have different efficacy for a greater reason, a group should have been envisaged that had both interventions, perhaps with a lower caloric restriction (which seems to be on average 500kcal, in my opinion too high)

Author Response

(The authors gave the same response as above.)

Round 2

Reviewer 1 Report

The authors have adequately addressed all the reviewer's issues and thoroughly revised the manuscript. This revision has significantly improved the manuscript.

Author Response

Many thanks for the reviewer's comments.

Reviewer 2 Report

The authors replied to my clarifications, but many points cannot be changed, so a paragraph with the limitations of the study should be added:

- Lack of body composition assessment
- High caloric deficit and difficult to sustain.
- Food diary as evaluation and operated for 4 days.
- Lack of control and a group that includes exercise and diet
- Indicative dietary program and evaluated afterward and not personalized.

Author Response

Many thanks for the reviewer's comments, please see attached file for our response
